# Terahertz chiral photonic-crystal cavities for Dirac gap engineering in graphene

Fuyang Tay [1,2] ✉, Stephen Sanders [1], Andrey Baydin [1,3,4], Zhigang Song [5], Davis M. Welakuh[6], Alessandro Alabastri [1,3,4], Vasil Rokaj[7,8,9], Ceren B. Dag [7,8,10] & Junichiro Kono [1,3,4,11,12] ✉

Strong coupling between matter and vacuum electromagnetic fields in a cavity can induce novel quantum phases in thermal equilibrium via symmetry breaking. Particularly intriguing is the coupling with circularly polarized cavity fields, which can break time-reversal symmetry (TRS) and lead to topological bands. This has spurred significant interest in developing chiral cavities that feature broken TRS, especially in the terahertz (THz) frequency range, where various large-oscillator-strength resonances exist. Here, we present a design for high-quality-factor THz chiral photonic-crystal cavities (PCCs) that achieve broken TRS using a magnetoplasma in a lightly doped semiconductor. We incorporate ab initio density functional theory calculations into the derived microscopic model, allowing a realistic estimate of the vacuum-induced gap in graphene when coupled to our chiral cavity. Our calculations show an enhancement in the light–matter interaction due to Dirac nodes and predict an energy gap on the order of 1 meV. The THz chiral PCCs offer a promising platform for exploring cavity-dressed condensed matter with broken TRS.

The strong coupling of condensed matter with radiation inside a photonic cavity has recently attracted considerable attention due to the possibility that unusual new quantum phases may arise in thermal equilibrium without an external driving field. Namely, a certain matter resonance can be strongly coupled, or dressed, by vacuum electromagnetic fields, or virtual photons, that surround the matter in the cavity[1–4]. Recent experiments on condensed matter in cavities have revealed new effects such as the breakdown of the topological protection of the quantum Hall effect[5,6] and thermal control of the metal-to-insulator transition in a charge-density-wave system[7]. One approach toward the creation of a new ground state in a material is by breaking a specific material symmetry with light. For example, circularly polarized light with finite angular momentum can break time-reversal symmetry

(TRS), resulting in a gap opening and the formation of topological Floquet bands in graphene[8]. Recent theoretical studies suggest that coupling a material with a circularly polarized vacuum field inside a chiral cavity can achieve band structure engineering similar to Floquet physics[1,9,10]. Therefore, a robust chiral cavity design that can be used to break TRS in a material is desired.

Past studies have focused on chiral artificial materials that only break mirror symmetry, including photonic crystals[11–15] and metamaterials and plasmonic nanostructures[16–20], by using specific geometrical designs. In recent years, there has been discussion and realization of chiral cavities with broken TRS. First, a chiral cavity consisting of two Faraday mirrors, each integrating a Faraday rotator and a regular mirror, has been proposed[1,21]. The Faraday rotator

[1]Department of Electrical and Computer Engineering, Rice University, Houston, TX, USA. [2]Applied Physics Graduate Program, Smalley–Curl Institute, Rice University, Houston, TX, USA. [3]Smalley–Curl Institute, Rice University, Houston, TX, USA. [4]Rice Advanced Materials Institute, Rice University, Houston, TX, USA. [5]John A. Paulson School of Engineering and Applied Sciences, Harvard University, Cambridge, MA, USA. [6]Max Planck Institute for the Structure and Dynamics of Matter, Hamburg, Germany. [7]Department of Physics, Harvard University, Cambridge, MA, USA. [8]ITAMP, Harvard-Smithsonian Center for Astrophysics, Cambridge, MA, USA. [9]Department of Physics, Villanova University, Villanova, PA, USA. [10]Department of Physics, Indiana University, Bloomington, IN, USA. [11]Department of Physics and Astronomy, Rice University, Houston, TX, USA. [12]Department of Materials Science and NanoEngineering, Rice University, Houston, TX, USA. ✉e-mail: fuyang.tay@columbia.edu; kono@rice.edu

induces a phase difference between light of opposite handedness, causing the nodes and antinodes of their standing waves to occupy different positions within the cavity. Ideally, when the phase difference between the reflected light of opposite handedness equals an odd multiple of $\pi/2$, the antinodes of one chiral standing wave coincide with the nodes of the other, resulting in an enhanced electric field with a single-handedness[21]. Similar designs, including Fabry–Pérot and Bragg cavities incorporating Faraday rotation materials, have been explored earlier[22–24] to enhance magneto-optic effects and achieve unidirectional transmission. However, the reported Faraday rotation angles have generally been small, and no dedicated studies have specifically aimed at realizing a fully chiral cavity. More recently, chiral cavities with broken TRS have been demonstrated based on different mechanisms[25–27]. Suárez-Forero et al. have realized a chiral cavity consisting of two MoSe$_2$ mirrors that exhibit spin-selective reflection in a high magnetic field[27]. The Fabry–Pérot modes in the visible range were made nondegenerate between left-circular (LCP) and right-circular polarizations (RCP) by the applied magnetic field. Andberger et al. and Aupiais et al. have developed chiral plasmonic cavities that operate in the terahertz (THz) frequency range by coupling plasmonic resonators with the cyclotron resonance of a two-dimensional electron gas in GaAs[25] or a magnetoplasma in bulk InSb[26]. The quality ($Q$) factor of the fabricated cavity remained low, and the degree of chirality, or ellipticity, was not perfect and spatially uniform.

Here, we describe a scheme for designing a THz chiral photonic-crystal cavity (PCC) with broken TRS by using a magnetoplasma in lightly doped InSb. Due to InSb's low effective mass, its cyclotron resonance frequency falls within the THz range under a small applied

magnetic field. The nonreciprocal nature of THz transmission through the magnetoplasma, arising from free-carrier cyclotron resonance, enables selective absorption of circularly polarized light propagating within the PCCs. The carrier density is chosen so that the dielectric constant of InSb closely matches that of intrinsic silicon for the opposite handedness, thereby maintaining the high $Q$ factor of the PCC. Through simulations with various designs, we analyzed the transmittance spectra, mode, and ellipticity profiles of the chiral PCCs. In an optimized structure, we achieved a THz chiral PCC with a chiral mode at approximately 0.42 THz in a low magnetic field of around 0.2 T. The $Q$ factor of the cavity remains high (>400), with a uniformly circularly polarized cavity electric field in the lateral plane of the central defect layer surfaces, which is ideal for integrating two-dimensional materials and thin film samples. Furthermore, we utilize ab-initio density functional theory (DFT) to calculate the transition matrix elements of monolayer graphene, which determines the light–matter interaction strength, as we analytically show within a cavity QED model. This demonstrates a significant enhancement of the light–matter interaction at the Dirac points. Hence, our theoretical approach brings together the exact cavity-field profile through first-principle simulations of Maxwell's equations, and the transitions between graphene bands obtained from ab-initio DFT, in a cavity QED tight-binding model of graphene, see Fig. 1a. This comprehensive approach enables a realistic estimation of the cavity-induced topological gap opening at the Dirac nodes, which reaches approximately 1 meV. Importantly, the cavity-induced topological gap can be engineered through further reduction of the cavity mode volume.

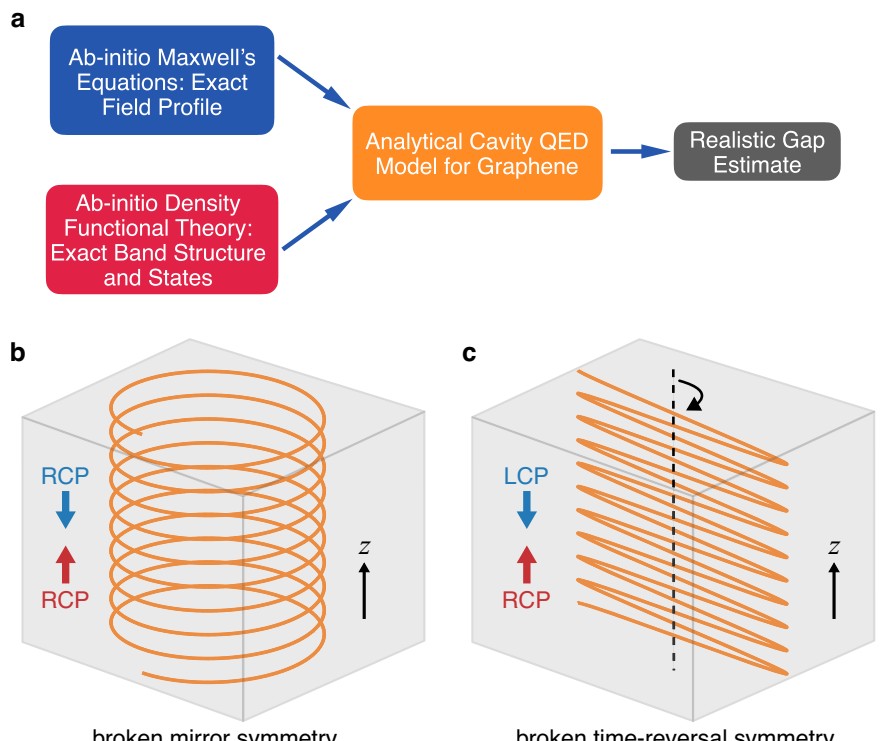

**Fig. 1 | Schematic diagram illustrating Dirac gap estimation and standing wave formation in a chiral slab with two types of chirality−broken mirror symmetry vs broken time-reversal symmetry. a** Combining Maxwell's equations with density functional theory-based ab-initio calculations in a cavity quantum electrodynamics (QED) model for graphene, enabling a realistic estimation of the Dirac gap induced by chiral cavity modes. **b, c** Chiral standing waves (orange trace) occur when circularly polarized light propagates in opposite directions along the *z*-axis (red and blue arrows) within a slab (gray box). In the case of broken mirror symmetry (**b**), the counter-propagating lights have the same handedness, forming a helical standing wave along the *z*-axis with a sinusoidally varying amplitude over time. By contrast, for broken time-reversal symmetry (**c**), the counter-propagating lights have opposite handedness, resulting in a sinusoidal standing wave along the *z*-axis with its polarization that rotates constantly over time. Notably, only in (**c**) is the electric field circularly polarized at all *z*-positions. RCP right-handed circular polarization, LCP left-handed circular polarization.

## Results

### Distinctions between chiral cavity types

Here, we provide a concise overview of the distinction between chiral cavities with broken mirror symmetry and those with broken TRS. Chiral matter exhibits different optical responses to LCP and RCP light. Chirality can arise from breaking either mirror symmetry (e.g., via spiral structures) or TRS (e.g., through an external magnetic field). The characteristics of standing waves formed within a chiral system depend on the type of broken symmetry.

In systems lacking mirror symmetry but preserving TRS, the optical response to circularly polarized light is reciprocal. Specifically, the responses to LCP and RCP light remain unchanged when the direction of propagation is reversed[28]. For example, in a chiral slab that supports only the propagation of RCP light in both directions, RCP waves traveling in opposite directions along the $z$-axis form a helical standing wave inside the slab, as illustrated in Fig. 1b. The polarization of the helical standing wave rotates along the $z$ direction, while the amplitude varies sinusoidally with time $t$, given by $\mathbf{E}_{\text{standing}} \propto \cos(\omega t)[\mathbf{e}_x \cos(kz) \pm \mathbf{e}_y \sin(kz)]$[17,21], where $k$ is the wave vector and $\omega$ is the angular frequency. Note that the electric field of this standing wave remains linearly polarized at any given $z$.

By contrast, chiral systems with broken TRS exhibit nonreciprocal behavior, where the roles of LCP and RCP light reverse when the propagation direction is flipped, while the magnetic field direction remains unchanged. For example, in a chiral slab that supports only forward RCP and backward LCP light propagation, the resulting standing wave becomes a static cosine wave along the $z$ direction, with its polarization rotating over $t$, as depicted in Fig. 1c. This wave can be described by $\mathbf{E}_{\text{standing}} \propto \cos(kz)[\mathbf{e}_x \sin(\omega t) \pm \mathbf{e}_y \cos(\omega t)]$[17,21]. In this case, the electric field of the standing wave is circularly polarized at any $z$.

In summary, the nature of standing waves in a chiral system depends on the type of broken symmetry due to distinct helicity transformations upon reflection.

### Time-reversal symmetry in hybrid photo-electronic systems

Here, we review how TRS is manifested in systems where charged particles are coupled to the quantized cavity fields. For a single electron in a periodic crystal potential coupled to light, the minimal coupling Hamiltonian[29,30] follows as,

$$\mathcal{H} = \frac{1}{2m}(i\hbar\nabla + e\mathbf{A})^2 + V_{\text{crys}}(\mathbf{r}) + \sum_{\lambda = x,y} \hbar\omega_{\text{cav}}\left(b_\lambda^\dagger b_\lambda + \frac{1}{2}\right), \quad (1)$$

with a single cavity mode $\omega_{\text{cav}} = c|\kappa_z|$ where $\kappa_z$ is the photon wave number along the $z$ direction, and $m$ is the bare electron mass. Under this choice, the polarizations of the cavity field are in the $(x, y)$ plane. The operators $b_\lambda^\dagger, b_\lambda$ are the creation and annihilation operators of the photon field satisfying bosonic commutation relations $[b_\lambda, b_{\lambda'}^\dagger] = \delta_{\lambda\lambda'}$, and $V_{\text{crys}}(\mathbf{r})$ is the crystal potential of graphene. For two-dimensional materials whose thickness is at the order of a single or a few atoms[31,32], the cavity field in the $z$ direction of our chiral PCC is taken to be uniform. Hence, the vector potential reads,

$$\mathbf{A} = \sqrt{\frac{\hbar}{\varepsilon_0 \mathcal{V} 2\omega_{\text{cav}}}} \sum_{\lambda = x,y} \mathbf{e}_\lambda \left(b_\lambda^\dagger + b_\lambda\right), \quad (2)$$

where $\mathcal{V}$ is the effective mode volume, $\mathbf{e}_x$ and $\mathbf{e}_y$ are the linear polarization vectors of light.

In classical physics, the momentum $\mathbf{p}$ of a particle and the classical vector potential $\mathbf{A}_{\text{cl}}(t)$ transform under time-reversal $\mathcal{T}$ as $\mathcal{T}(\mathbf{p}) = -\mathbf{p}$ and $\mathcal{T}(\mathbf{A}_{\text{cl}}) = -\mathbf{A}_{\text{cl}}$[33], such that a linearly polarized electric field $\mathbf{E}_{\text{cl}}(t)$ is invariant under TRS. These transformation rules must be preserved under quantization leading to $\mathcal{T}(-i\hbar\nabla) = i\hbar\nabla$ and $\mathcal{T}(\mathbf{A}) = -\mathbf{A}$. Under this transformation $\{b_{x,y}, b_{x,y}^\dagger\}$ have to satisfy

$\mathcal{T}(b_{x,y}) = -b_{x,y}$ and $\mathcal{T}(b_{x,y}^\dagger) = -b_{x,y}^\dagger$. With the use of these relations, we can find that the photon number operator $b_\lambda^\dagger b_\lambda$ is invariant under TRS $\mathcal{T}(b_\lambda^\dagger b_\lambda) = b_\lambda^\dagger b_\lambda$ for linearly polarized light. By combining all transformation rules, we find that the minimal coupling Hamiltonian for linearly polarized light is invariant under TRS $\mathcal{T}(\mathcal{H}) = \mathcal{H}$.

To discuss TRS under circular polarization, we make a basis change in the polarization vectors from $\mathbf{e}_x$ and $\mathbf{e}_y$ to left $\mathbf{e}_L = (1, i)/\sqrt{2}$ and right $\mathbf{e}_R = (1, -i)/\sqrt{2}$ circular polarizations through the expressions $\mathbf{e}_x = (\mathbf{e}_R + \mathbf{e}_L)/\sqrt{2}$ and $\mathbf{e}_y = i(\mathbf{e}_R - \mathbf{e}_L)/\sqrt{2}$[34]. Then the photon field takes the form

$$\mathbf{A} = \sqrt{\frac{\hbar}{\varepsilon_0 \mathcal{V} 2\omega_{\text{cav}}}}\left[\mathbf{e}_R b_L + \mathbf{e}_R b_R^\dagger + \mathbf{e}_L b_R + \mathbf{e}_L b_L^\dagger\right], \quad (3)$$

where we defined the photon operators for left and right circularly polarized photons as linear combinations of the linearly polarized ones, $b_L = \frac{1}{\sqrt{2}}\left(b_x + ib_y\right)$ and $b_R = \frac{1}{\sqrt{2}}\left(b_x - ib_y\right)$. It can be checked that the left and right-handed photon operators satisfy standard bosonic commutation relations $[b_L, b_L^\dagger] = [b_R, b_R^\dagger] = 1$, and are independent $[b_L, b_R^\dagger] = 0$. Using the definition of the left and right-polarized photon operators, we find their transformation under TRS to be

$$\mathcal{T}(b_{L,R}) = -b_{R,L} \text{ and } \mathcal{T}(b_{L,R}^\dagger) = -b_{R,L}^\dagger. \quad (4)$$

Thus, we see that TRS exchanges left and right photon operators (up to a minus). The same also holds for left and right polarization vectors $\mathcal{T}(\mathbf{e}_R) = \mathbf{e}_L$ and $\mathcal{T}(\mathbf{e}_L) = \mathbf{e}_R$. Hence, expectantly $\mathcal{T}(\mathbf{A}) = -\mathbf{A}$ still holds, while the energy of the photon field in terms of left and right circularly polarized operators takes the standard form $\sum_{\lambda = x,y} \hbar\omega_{\text{cav}}(b_\lambda^\dagger b_\lambda + \frac{1}{2}) = \sum_{\lambda = L,R} \hbar\omega_{\text{cav}}(b_\lambda^\dagger b_\lambda + \frac{1}{2})$, being invariant under TRS, too. If the right circularly polarized photons are completely absorbed by the magnetoplasma in InSb, the cavity field would read

$$\mathbf{A} = \mathbf{A}_L = \sqrt{\frac{\hbar}{\varepsilon_0 \mathcal{V} 2\omega_{\text{cav}}}}\left[\mathbf{e}_R b_L + \mathbf{e}_L b_L^\dagger\right]. \quad (5)$$

It is straightforward to see $\mathcal{T}(\mathbf{A}_L) = -\mathbf{A}_R$ breaking TRS. Let us note that even when the absorption of one polarization by InSb is imperfect, we would still have a field of the form $\mathbf{A}' = \alpha_L \mathbf{A}_L + \alpha_R \mathbf{A}_R$ where $\alpha_R \neq \alpha_L$, and hence the TRS is still broken, $\mathcal{T}(\mathbf{A}') \neq -\mathbf{A}'$[10].

### Chiral 1D-PCC: design principle and optimum structure

Let us first consider a conventional 1D-PCC composed of five silicon layers (i–v) separated by air, as depicted in Fig. 2a. The refractive indices of the layers at THz frequencies are $n_{\text{Si}} = 3.42$ and $n_{\text{air}} = 1$. The layer thicknesses are $d_{\text{Si}} = 50\,\mu m$ and $d_{\text{air}} = 198\,\mu m$. Layer (iii) is twice as thick as the other layers, acting as a defect layer in the 1D-PCC. We used COMSOL Multiphysics 6.2 to calculate the normalized power, $P$, of transmitted THz radiation for different polarizations (+for LCP and − for RCP), as shown in Fig. 2b. The incident THz beam is linearly polarized in the $x$ direction. Figure 2b reveals a high-$Q$ defect mode ($Q = 493$) at $\omega_{\text{cav}}/2\pi = 0.423$ THz within the photonic band gap between around 0.25 THz and 0.55 THz.

As the conventional 1D-PCC is not chiral, the transmitted light remains linearly polarized in the $x$ direction ($P_x \approx 1$, $P_y = 0$, and $P_+ = P_- \approx 0.5$). Figure 2c displays the spatial profiles of the cavity electric field in the − polarization, $|E_-(z)| = |(E_x(z) - iE_y(z))/\sqrt{2}|$, and ellipticity, $\eta(z) = \frac{|E_-(z)| - |E_+(z)|}{|E_-(z)| + |E_+(z)|}$, at the peak frequency of the 1D-PCC. The electric field is linearly (circularly) polarized if $\eta(z) = 0$ ($\eta(z) = \pm 1$). At the mode frequency, the electric field is highly localized near the surfaces of the defect layer, $z_{\text{max}}$, with $\eta = 0$ across the cavity. This conventional 1D-PCC design has been utilized to achieve ultrastrong coupling by placing an ultrahigh-mobility two-dimensional electron gas at $z_{\text{max}}$[35,36].

Next, we consider replacing specific silicon layers in the 1D-PCC with lightly doped $n$-InSb layers. Lightly doped $n$-InSb contains a low-

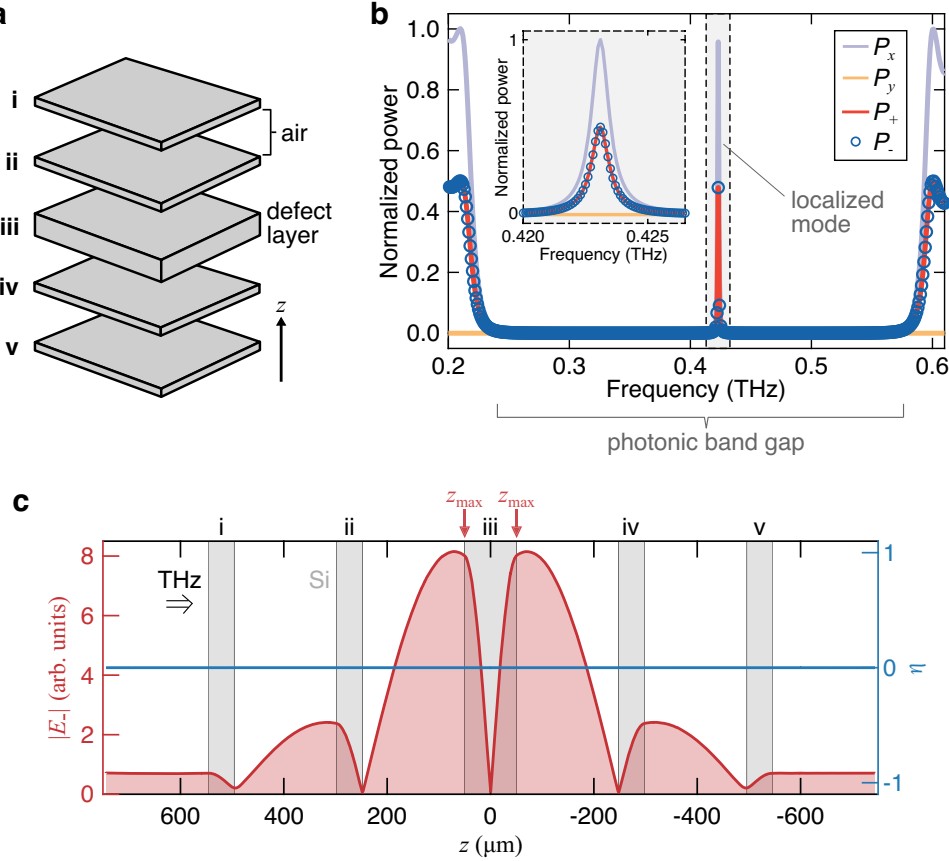

**Fig. 2 | A conventional 1D-PCC. a** Schematic diagram of the 1D-PCC. It consists of five layers, labeled layers (i–v), separated by a 198-$\mu$m-thick air gap. Layer (iii) is the defect layer with a thickness of 100 $\mu$m, which is thicker than the other layers (50 $\mu$m). **b** Normalized power ($P$) spectra of transmitted light for different polarizations, with $+$ denoting LCP and $-$ denoting RCP. The incident light is linearly polarized along the $x$-axis. The inset shows a magnified view of the spectrum. **c** The spatial profiles of the circularly polarized electric field ($|E_-|$) and ellipticity ($\eta$) at the cavity mode frequency, shown by the red and blue traces, respectively. The gray rectangles represent the positions of the silicon layers. Terahertz (THz) light is incident from the left side. $z_{max}$ denotes the surfaces of the defect layer where the electric field reaches its maximum.

density plasma with a plasma frequency in the THz frequency range, and a magnetoplasma with nonreciprocity properties is formed when an external magnetic field, $B$, is applied[37–41]. The complex permittivity of the magnetoplasma at $B$ is given by a gyrotropic permittivity tensor[42–46]

$$\tilde{\varepsilon} = \begin{pmatrix} \varepsilon_{xx}(\omega) & \varepsilon_{xy}(\omega) & 0 \\ -\varepsilon_{xy}(\omega) & \varepsilon_{xx}(\omega) & 0 \\ 0 & 0 & \varepsilon_{zz} \end{pmatrix}, \quad (6)$$

with

$$\varepsilon_{xx}(\omega) = \varepsilon_{bg} - \frac{\omega_p^2(\omega - i\gamma)}{\omega[(\omega - i\gamma)^2 - \omega_c^2]}, \quad (7)$$

$$\varepsilon_{xy}(\omega) = \frac{-i\omega_p^2\omega_c}{\omega[(\omega - i\gamma)^2 - \omega_c^2]}, \quad (8)$$

$$\varepsilon_{zz} = \varepsilon_{bg} - \frac{\omega_p^2}{\omega(\omega - i\gamma)}, \quad (9)$$

where $\varepsilon_{bg}$ represents the background permittivity, $\omega_p = \sqrt{n_e e^2/(\varepsilon_0 m_{eff})}$ is the plasma frequency, $n_e$ denotes the electron density, $e$ is the electronic charge, $\varepsilon_0$ is the permittivity of free space,

$m_{eff}$ is the effective mass of the electrons, $\gamma$ is the scattering rate, and $\omega_c = eB/m_{eff}$ is the cyclotron frequency. The magnetoplasma parameters we used in this study are similar to those reported in previous experimental studies[47–49]: $n_e = 2.3 \times 10^{14}$ cm$^{-3}$, $m_{eff} = 0.014\,m$ (where $m = 9.11 \times 10^{-31}$ kg is the free electron mass in vacuum), and $\gamma = 1.5 \times 10^{11}$ rad/s.

Figure 3a displays the complex permittivity of the magnetoplasma in the circular basis, $\tilde{\varepsilon}_\pm = \varepsilon'_\pm - i\varepsilon''_\pm$, at $B = 0.212$ T, where $\omega_{cav} = \omega_c$. $\tilde{\varepsilon}_+$ shows a Lorentzian peak at $\omega_c$, indicating absorption of LCP radiation. By contrast, the magnetoplasma transmits RCP light at $\omega_c$ because $\varepsilon'_- > 0$ and $\varepsilon''_- \approx 0$. Due to the breaking of TRS by the external $B$, the roles of LCP and RCP light interchange when the propagation direction is reversed (Fig. 3b). Specifically, RCP light is absorbed, while LCP light can pass through the InSb layer in this case.

The nonreciprocal nature of transmission through the magnetoplasma allows an InSb layer in $B$ to selectively absorb specific circularly polarized light depending on the propagation direction. This property of InSb can be utilized in a 1D-PCC design, because it can maintain counterpropagating waves with opposite handedness inside PCCs. Therefore, the defect mode of the PCC becomes chiral when $\omega_{cav} = \omega_c$. The low $m_{eff}$ in InSb is advantageous for creating THz chiral PCCs because it requires only a small $B$ to shift $\omega_c$ into the THz range. Furthermore, from Fig. 3a, we can see that $\varepsilon'_- \simeq 12.3$ at $\omega_{cav}$, which is comparable to the permittivity of silicon. Therefore, replacing silicon layers with InSb layers does not significantly alter the photonic band gap and the frequency of the defect mode of the PCCs. Note that $\varepsilon'_- < 0$

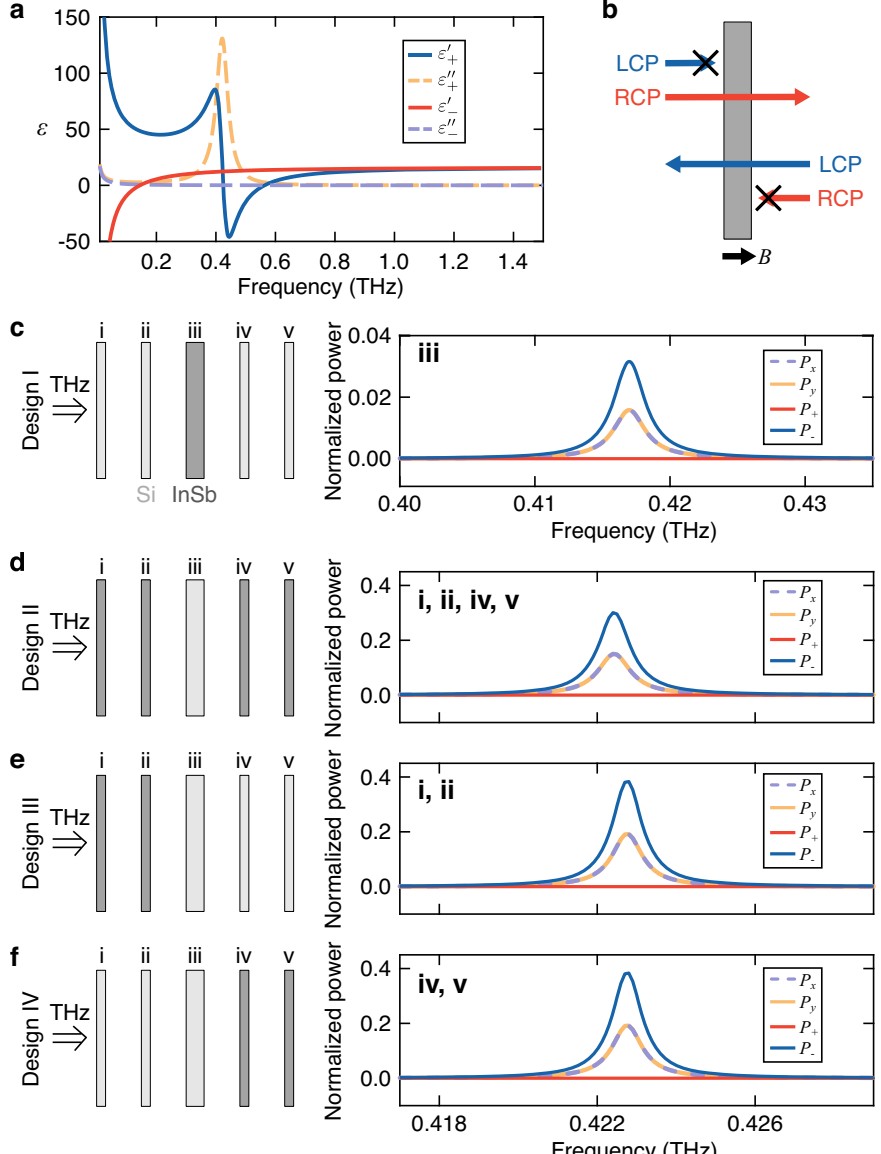

**Fig. 3 | Nonreciprocal THz magnetoplasma in lightly doped InSb and its incorporation into a 1D-PCC. a** Complex permittivity, $\tilde{\varepsilon}_\pm = \varepsilon'_\pm - i\varepsilon''_\pm$, of the magnetoplasma in the circular basis at a magnetic field of $B = 0.212$ T, where light propagates in the same direction as $B$ (Faraday geometry). **b** Schematic diagram illustrating nonreciprocal transmission of circularly polarized light through the magnetoplasma. Blue and red arrows indicate the propagation direction of LCP and RCP light, respectively. When the propagation direction aligns with (opposes) the $B$ direction, denoted by the black arrow, only RCP (LCP) light can transmit through the InSb layer (gray rectangle). **c**–**f** Normalized power ($P$) spectra for transmitted light for different polarizations; + corresponds to LCP and − corresponds to RCP. The incident THz beam is linearly polarized in the $x$ direction. The left panels illustrate the configuration of four different designs (I–IV), with light and dark rectangles representing Si and InSb layers, respectively. Labels (i–v) in the right panels denote the layers that are InSb. RCP right-handed circular polarization, LCP left-handed circular polarization.

when $n_e$ is higher, imposing a limit on the maximum $n_e$ feasible for achieving a chiral 1D-PCC.

In this study, we consider four different configurations of chiral 1D-PCCs, as shown in Fig. 3c–f.

- Design I: only layer (iii) (the defect layer) is InSb;
- Design II: all layers, except layer (iii), are InSb;
- Design III: only layers (i) and (ii) are InSb;
- Design IV: only layers (iv) and (v) are InSb.

First, we investigate the structure in Design I (Fig. 3c). The transmittance spectrum of this structure has been previously studied[50–52]. Although the incident light is linearly polarized in the $x$ direction, the transmitted light becomes fully circularly polarized as $P_+ \approx 0$, $P_- \approx 2P_x = 2P_y$. However, the amplitude of the defect mode in the

transmittance spectrum significantly decreases (<0.04), and the peak broadens ($Q = 146$). The peak frequency slightly decreases as well. Next, for the structure in Design II, the transmitted light remains circularly polarized as $P_+ \approx 0$ and $P_- \approx 2P_x = 2P_y$, as shown in Fig. 3d. However, the amplitude of the peak for $P_-$ is much higher ($\simeq 0.3$). The $Q$ factor of the cavity is approximately 427, which is comparable to that of a conventional 1D-PCC. Finally, we considered structures where only the first two layers (Design III) or the last two layers (Design IV) are replaced by InSb. The transmittance spectra are identical for both cases, as shown in Fig. 3e, f. The amplitude of the peaks increases further, and the $Q$ factor reaches 458, while the transmitted light remains circularly polarized.

In addition to achieving circular dichroism, our primary goal is to enhance circularly polarized vacuum electric fields. To examine this,

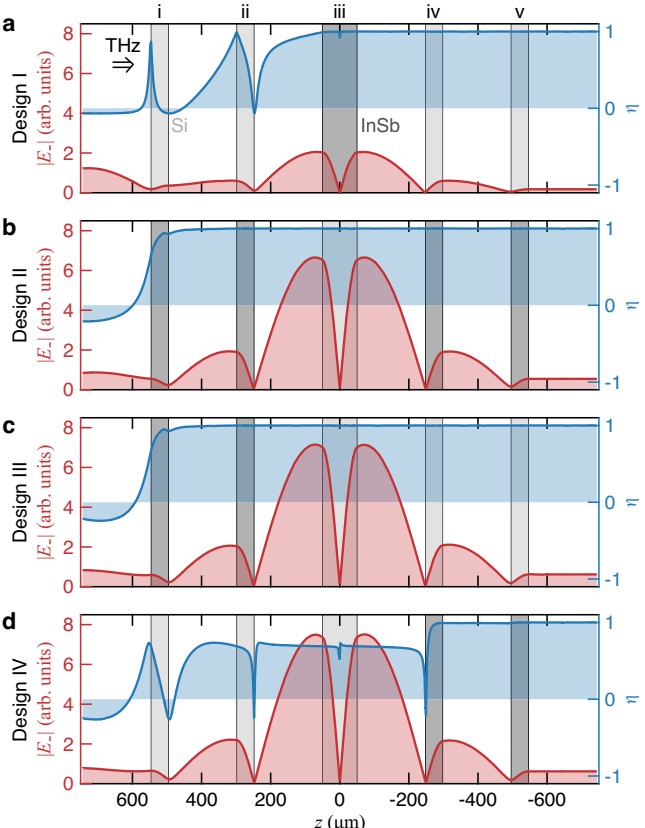

**Fig. 4 | Cavity electric field profiles of THz chiral 1D-PCCs. a, d** Mode and ellipticity profiles of chiral cavities in four different designs (I–IV). The red and blue traces represent the circularly polarized cavity electric field $|E_-|$ and ellipticity, $\eta$, respectively. The light and dark rectangles represent Si and InSb layers, respectively, that compose the 1D-PCCs. The following layers are InSb in the different designs: **a** iii; **b** i, ii, and iv; **c** i and ii; **d** iv and v. The THz beam is incident from the left side and linearly polarized in the $x$ direction.

we investigate the spatial profiles of $|E_-(z)|$ and $\eta(z)$ in four different designs, as shown in Fig. 4; the incident light is linearly polarized in the $x$ direction. Note that all four designs exhibit $|E_-(z)|$ profiles similar to those of conventional 1D-PCCs, with maximum $E_-$ occurring at the surfaces of the defect layer. However, the transmitted power and the cavity electric field of the PCCs are particularly sensitive to losses in the defect layer compared to losses in other layers; see Supplementary Section 2 and Supplementary Fig. 2 and 3. Thus, the amplitude of $|E_-(z)|$ in Design I significantly decreases (Fig. 4a). By contrast, the peak amplitude of $|E_-(z)|$ in Design II remains high (Fig. 4b). Notably, despite the defect layer being nonmagnetic in Design II, the value of $\eta(z)$ at $z_{max}$ (the defect layer surface) still reaches 1. This configuration differs from the conventional structure explored in previous studies, which typically incorporates a magnetic defect layer within a photonic crystal[50–52].

The transmittance spectra suggest that the $Q$ factor of the chiral mode in Designs III and IV is higher (Fig. 4c, d). However, $\eta(z) < 1$ at $z_{max}$ if light passes through the defect layer prior to the InSb layers (Fig. 4d). Given that vacuum fields can approach from either side of the cavity, we conclude that the optimized chiral 1D-PCC design, which has broken TRS and features enhanced chiral vacuum fields, is Design II, which is depicted in Fig. 4b: a five-layer PCC composed of air and InSb layers with a silicon defect layer in the middle.

The presence of a magnetoplasma in InSb causes significant changes in both the transmittance spectra and mode profiles of our chiral PCCs as $B$ is varied, see Supplementary Section 4 and

Supplementary Fig. 4. At $B \neq 0.212$ T, $\eta(z) < 1$ at $z_{max}$ as $\omega_c$ is detuned from $\omega_{cav}$. Furthermore, the handedness of the chiral resonance reverses when the sign of $B$ is inverted (Supplementary Fig. 5).

Finally, we calculated the spatial profile of vacuum fluctuations of the chiral mode in the chiral PCCs by normalizing the mode profiles obtained from numerical simulations with circularly polarized incident THz beams; see Supplementary Section 6. The circularly polarized vacuum field strength, $|E_{vac,-}|$ at $z_{max}$ is approximately $1.35 \times 10^{-4}/\sqrt{S}$ V/m, where $S$ denotes the surface area of the cavity (Supplementary Fig. 7). In addition, our approach can be utilized to realize chiral Tamm cavities for reflection measurements. A Tamm cavity is a combination of a photonic crystal and a metallic mirror[53]. The transmittance spectra and mode profiles of a chiral Tamm cavity, which consists of two InSb layers and a silicon layer coated with gold, are discussed in Supplementary Section 5 and Supplementary Fig. 6. The amplitude of $|E_{vac,-}|$ increases to $1.92 \times 10^{-4}/\sqrt{S}$ V/m at $z_{max}$, while $\eta(z_{max}) \simeq 1$ is achieved (Supplementary Fig. 7).

## Advantages of our chiral PCCs

Our chiral PCC designs offer several advantages compared to previously reported chiral cavities with broken TRS[25–27]. First, the required $B$ for our chiral PCCs is relatively low (~0.2 T) due to the low $m_{eff}$ of electrons in InSb. This minimizes changes in the material properties induced solely by the external $B$. Second, the cavity electric field is uniformly circularly polarized in the $xy$ plane. Third, the $Q$ factors of the chiral PCCs remain high ($Q > 400$). Fourth, the defect layer in the optimum design (Design II), which serves as the substrate for materials embedded within the cavity, is made of silicon. This choice helps eliminate potential parasitic effects that could arise if the material were placed on a metallic substrate. Moreover, the mode volumes of the chiral PCCs can potentially be reduced by integrating air slots or bowtie structures into the multilayer design[54] or by coupling metasurface resonators with the chiral PCC[55,56]. Two-dimensional materials and thin film samples can be conveniently embedded in the chiral PCCs to examine the effects of chiral vacuum fields on material properties in the vacuum state. Below, we estimate the gap at the Dirac nodes of monolayer graphene placed inside our chiral PCC.

## Tight-binding model with density functional theory and Dirac gap estimation

Next, we derive the tight-binding Hamiltonian for graphene coupled to our single polarization chiral PCC. The study of monolayer graphene strongly coupled to chiral cavity modes has attracted considerable interest in recent years[9,10,57–59]. Here, we will follow the approach developed in ref. 10 and extend it further by including ab initio calculations for the graphene monolayer. Expanding the covariant kinetic energy in the minimal-coupling Hamiltonian in Eq. (1), we have

$$\mathcal{H} = \underbrace{-\frac{\hbar^2}{2m}\nabla^2 + V_{crys}(\mathbf{r})}_{\text{Matter: } \mathcal{H}_m} + \underbrace{\frac{ie\hbar}{m}\mathbf{A}\cdot\nabla}_{\text{Photon-Matter: } \mathcal{H}_{pm}} + \underbrace{\frac{e^2}{2m}\hat{\mathbf{A}}^2 + \hbar\omega_{cav}\left(b_L^\dagger b_L + \frac{1}{2}\right)}_{\text{Photonic: } \mathcal{H}_p}.$$

$$(10)$$

where the external potential is periodic under Bravais lattice translations $V_{crys}(\mathbf{r} + \mathbf{R_j}) = V_{crys}(\mathbf{r})$ with $\mathbf{R_j}$ being the Bravais lattice vectors[60]. Substituting the expression for the vector potential $\mathbf{A}_L$ and introducing the diamagnetic frequency $\omega_D = \sqrt{\frac{e^2}{m\varepsilon_0 \mathcal{V}}}$, the photonic Hamiltonian $\mathcal{H}_p$ takes the form

$$\mathcal{H}_p = \hbar\omega_{cav}\left(b_L^\dagger b_L + \frac{1}{2}\right) + \frac{\hbar\omega_D^2}{4\omega_{cav}}\left(b_L + b_L^\dagger\right)^2. \quad (11)$$

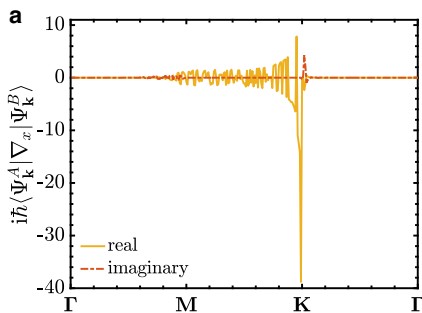
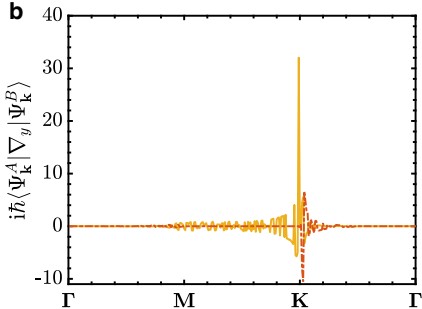

**Fig. 5 | Transition matrix elements for graphene. a, b** The matrix elements $i\hbar\langle\Psi_{\mathbf{k}}^A|\nabla|\Psi_{\mathbf{k}}^B\rangle$ in atomic units computed with density functional theory and the tight-binding model of graphene in (**a**) $x$-direction and (**b**) $y$-direction with respect to Brillouin zone momenta.

Now $\mathcal{H}_p$ can be brought into diagonal form by introducing a new set of bosonic operators: $a_L^\dagger$ and $a_L$

$$a_L = \frac{1}{2\sqrt{\omega_{\text{cav}}\omega}}\left[b_L(\omega+\omega_{\text{cav}})+b_L^\dagger(\omega-\omega_{\text{cav}})\right] \text{ with } \omega = \sqrt{\omega_{\text{cav}}^2+\omega_D^2}. \quad (12)$$

The frequency $\omega$ is the dressed cavity frequency, which depends on the bare cavity frequency $\omega_{\text{cav}}$ and the diamagnetic shift $\omega_D$[61], and the operators $a_L, a_L^\dagger$ satisfy bosonic commutation relations $[a_L, a_L^\dagger]=1$. Then $\mathcal{H}_p$ and $\mathbf{A}_L$ can be simply written as

$$\mathcal{H}_p = \hbar\omega\left(a_L^\dagger a_L+\frac{1}{2}\right), \quad \mathbf{A}_L = \left(\frac{\hbar}{\varepsilon_0\mathcal{V}}\right)^{\frac{1}{2}}\frac{1}{\sqrt{2\omega}}\left(\mathbf{e}_R a_L+\mathbf{e}_L a_L^\dagger\right). \quad (13)$$

Graphene has two sublattices, $A$ and $B$, and as a consequence, the tight-binding ansatz wavefunction consists of two components, one for each sublattice[62],

$$\Psi_{\mathbf{k}}(\mathbf{r}) = a_{\mathbf{k}}\Psi_{\mathbf{k}}^A(\mathbf{r})+b_{\mathbf{k}}\Psi_{\mathbf{k}}^B(\mathbf{r}) = \sum_{\mathbf{j}}e^{i\mathbf{k}\cdot\mathbf{R}_{\mathbf{j}}}\left[a_{\mathbf{k}}\phi_A(\mathbf{r}-\mathbf{R}_{\mathbf{j}})+b_{\mathbf{k}}\phi_B(\mathbf{r}-\mathbf{R}_{\mathbf{j}}^B)\right], \quad (14)$$

where $\mathbf{R}_{\mathbf{j}}=j_1\mathbf{a}_1+j_2\mathbf{a}_2$ are the Bravais vectors of sublattice $A$ with $\mathbf{a}_1 = \mathbf{e}_x\alpha\sqrt{3}/2+\mathbf{e}_y 3\alpha/2$ and $\mathbf{a}_2 = -\mathbf{e}_x\alpha\sqrt{3}/2+\mathbf{e}_y 3\alpha/2$, where $\alpha$ is the graphene lattice constant. The Bravais vectors for the sublattice $B$ are $\mathbf{R}_{\mathbf{j}}^B = \mathbf{R}_{\mathbf{j}}+\boldsymbol{\delta}_3$ with $\boldsymbol{\delta}_3 = -\alpha\mathbf{e}_y$. To derive the tight-binding Hamiltonian, we apply $\mathcal{H}$ to Eq. (14)

$$\mathcal{H}_{\mathbf{k}} = \begin{pmatrix} \langle\Psi_{\mathbf{k}}^A|\mathcal{H}|\Psi_{\mathbf{k}}^A\rangle & \langle\Psi_{\mathbf{k}}^A|\mathcal{H}|\Psi_{\mathbf{k}}^B\rangle \\ \langle\Psi_{\mathbf{k}}^A|\mathcal{H}|\Psi_{\mathbf{k}}^B\rangle^* & \langle\Psi_{\mathbf{k}}^B|\mathcal{H}|\Psi_{\mathbf{k}}^B\rangle \end{pmatrix}. \quad (15)$$

The tight-binding Hamiltonian of graphene, $\langle\Psi_{\mathbf{k}}^A|\mathcal{H}_m|\Psi_{\mathbf{k}}^B\rangle = t[1+e^{i\mathbf{k}\cdot\mathbf{a}_1}+e^{i\mathbf{k}\cdot\mathbf{a}_2}]$, can be derived from the ansatz Eq. (14)[60,62] (see the "Methods" section for details). Here $t$ can be found by DFT as $t = 2.8$ eV[63]. Now we apply $\mathcal{H}_{pm}$ to Eq. (14)

$$\mathcal{H}_{\mathbf{k}}^{pm} = \begin{pmatrix} \langle\Psi_{\mathbf{k}}^A|\mathcal{H}_{pm}|\Psi_{\mathbf{k}}^A\rangle & \langle\Psi_{\mathbf{k}}^A|\mathcal{H}_{pm}|\Psi_{\mathbf{k}}^B\rangle \\ \langle\Psi_{\mathbf{k}}^A|\mathcal{H}_{pm}|\Psi_{\mathbf{k}}^B\rangle^* & \langle\Psi_{\mathbf{k}}^B|\mathcal{H}_{pm}|\Psi_{\mathbf{k}}^B\rangle \end{pmatrix}. \quad (16)$$

We neglect the diagonal terms, which results in tunneling between sites beyond nearest neighbors. Thus, we only need to compute $i\hbar\langle\Psi_{\mathbf{k}}^A|\nabla|\Psi_{\mathbf{k}}^B\rangle$. In ref. 10, an analytic perturbative approach with respect to the graphene lattice constant was followed for the derivation of the light–matter interaction term and its strength. Here, in order to make a more realistic prediction for cavity-induced Dirac gap, we incorporate ab-initio DFT simulations, which can provide the transition matrix elements between valence and conductions bands $-i\hbar\langle\Psi_{\mathbf{k}}^v|\nabla|\Psi_{\mathbf{k}}^c\rangle$ for graphene monolayer (see the "Methods" section for details), where $v$

and $c$ stand for valence and conduction bands, respectively. With a basis change from valence and conduction bands to A and B sublattices for the bare graphene, we find the relation

$$i\hbar\langle\Psi_{\mathbf{k}}^A|\nabla|\Psi_{\mathbf{k}}^B\rangle = -\sqrt{2}\frac{1+e^{i\mathbf{k}\cdot\mathbf{a}_1}+e^{i\mathbf{k}\cdot\mathbf{a}_2}}{|1+e^{i\mathbf{k}\cdot\mathbf{a}_1}+e^{i\mathbf{k}\cdot\mathbf{a}_2}|}\,\text{Re}\left[i\hbar\langle\Psi_{\mathbf{k}}^v|\nabla|\Psi_{\mathbf{k}}^c\rangle\right]. \quad (17)$$

We plot $i\hbar\langle\Psi_{\mathbf{k}}^A|\nabla|\Psi_{\mathbf{k}}^B\rangle$ in atomic units in Fig. 5, and observe more than an order of amplitude enhancement in $i\hbar\langle\Psi_{\mathbf{k}}^A|\nabla|\Psi_{\mathbf{k}}^B\rangle$ at the Dirac node compared to the rest of the Brillouin zone. Since $i\hbar\langle\Psi_{\mathbf{k}}^A|\nabla|\Psi_{\mathbf{k}}^B\rangle$ it dictates the light–matter interaction, we show below that such enhancement results in a sizable topological gap. To estimate the gap, we focus on the Dirac nodes, $\mathbf{K}$ and $\mathbf{K}'$. The relation above simplifies to $i\hbar\langle\Psi_{\mathbf{K}}^A|\nabla|\Psi_{\mathbf{K}}^B\rangle = -\sqrt{2}\,\text{Re}\left[i\hbar\langle\Psi_{\mathbf{K}}^v|\nabla|\Psi_{\mathbf{K}}^c\rangle\right]$ the Dirac nodes. Hence, the light–matter interaction Hamiltonian at the Dirac node reads,

$$\mathcal{H}_{\mathbf{K}}^{pm} = \frac{e}{m}\begin{pmatrix} 0 & \mathbf{A}_L\cdot i\hbar\langle\Psi_{\mathbf{K}}^A|\nabla|\Psi_{\mathbf{K}}^B\rangle \\ -\mathbf{A}_L\cdot i\hbar\langle\Psi_{\mathbf{K}}^A|\nabla|\Psi_{\mathbf{K}}^B\rangle^* & 0 \end{pmatrix}. \quad (18)$$

Remembering $\mathbf{A}_L$ in Eq. (13), we can define a light–matter coupling constant as

$$g = \frac{e}{m}\sqrt{\frac{2\pi}{\mathcal{V}\omega}}. \quad (19)$$

Effective cavity volume visible to graphene will be affected by the refractive index of the defect layer, $\mathcal{V} = \chi\left(\frac{2\pi c}{\omega_{\text{cav}}n_{\text{Si}}}\right)^3$ where $c$ is the speed of light, and $\chi$ is the light confinement parameter[5,64], and since there is confinement in only one direction in our PCC, $\chi = 0.1034$. Although this leads to a relatively small $g \sim 8\times10^{-8}$ a.u., the matrix element $\langle\Psi_{\mathbf{k}}^A|\nabla|\Psi_{\mathbf{k}}^B\rangle$ at the Dirac nodes enhances the light–matter interaction due to linear dispersion, leading to a gap of 0.88 meV. This gap strength can be further enhanced by engineering the light confinement within the chiral PCCs. One promising approach is to couple metasurface resonators with the chiral PCCs[55,56]. For example, one can achieve $\chi \sim 3.2\times10^{-4}$ with this approach, leading to $g = 1.4\times10^{-6}$ a.u. and a gap of ~2 meV. By increasing the cavity frequency to $\omega_{\text{cav}} = 2$ THz, one can obtain a gap on the order of 10 meV.

## Discussion

In this study, we proposed and analyzed chiral PCCs with broken TRS. Unlike conventional PCCs, one of the dielectric media in our chiral PCCs is replaced by a lightly doped semiconductor such as InSb. The cavity resonance becomes chiral when the PCC mode frequency $\omega_{\text{cav}}$ overlaps with the cyclotron frequency of the magnetoplasma in InSb, $\omega_c$. Thus, the required $B$ for the chiral PCCs depends on $m_{\text{eff}}$ of the magnetoplasma. Our findings indicate that the $Q$ factor and the confined electric field strength inside the cavity remain almost unchanged if the defect layer remains dielectric. The optimized design is photonic crystals formed by air and InSb with a silicon defect layer in the middle

(Design II). The ellipticity of the cavity electric field at the surfaces of the defect layer is approximately 1 and spatially uniform in the lateral direction.

The static magnetic field required by the chiral cavity also breaks TRS in graphene, leading to the formation of Landau levels. While both chiral photons and a static magnetic field break TRS, the resulting phenomena differ: chiral photons can induce the quantum anomalous Hall effect, whereas the magnetic field causes Landau quantization. At $B = 0.2$ T, for graphene with a high Fermi velocity, $v_F \sim 10^6$ m/s, the energy spacing between the Landau levels $\nu = 1$ and $\nu = 0$ is on the order of 10 meV. In this study, we neglect the magnetic field effects in graphene to simplify the analysis. The treatment of fully filled graphene Landau levels coupled to a cavity field differs fundamentally from the existing theories for GaAs Landau levels coupled to the cavity[6,35,36,46,65–67], due to the anharmonic gaps between graphene Landau levels. The form of the cavity-induced electronic interactions in graphene under a magnetic field, as well as whether the chiral cavity fields can modify Landau level spacings, e.g., lift the degeneracy of the $\nu = 0$ state, remain as some of the key questions to answer in the near future.

Within this theoretical treatment without a magnetic field, we estimated a cavity-induced gap of ~1 meV by combining different theoretical approaches, and importantly by utilizing the enhancement in light–matter interactions around the Dirac nodes. To experimentally detect this gap, the gap size needs to be comparable to, or larger, than both thermal and disorder broadening. Recent experiments have reported high-mobility graphene with low Landau-level disorder broadening of $\Gamma \sim 1$ meV[68,69], indicating that the predicted cavity-induced gap could be detected in cryogenic transport measurements, as its magnitude is comparable to $\Gamma$. Moreover, reducing the cavity mode volume or increasing the cavity mode frequency could further enhance the gap size. However, realizing a vacuum-induced topological Chern insulator in graphene[9,10] requires significant improvements in chiral cavity design to break TRS with no external magnetic field. The required magnetic field for our chiral cavity design is almost two orders of magnitude lower than previous schemes[25,27], bringing us closer to the aim of cavity-induced quantum anomalous Hall effect. A possible route to realizing magnetic-field-free chiral cavities with broken TRS is to replace the low-doped semiconductor with a material that exhibits a magnetic resonance at zero field, such as antiferromagnetic insulators. The low-loss chiral PCCs with broken TRS can also be applied for exploring chiral vacuum field-induced modifications in other materials[1] and chiral quantum optics[70,71].

## Methods
### Numerical simulations
The transmitted power spectra and electric field mode profiles of the cavities are computed using COMSOL Multiphysics 6.2. The simulation domain consists of a multilayer structure arranged as: perfectly matched layer (PML)/air/cavity/air/PML. Periodic boundary conditions are applied to the lateral (side) walls to model an infinite array, while scattering boundary conditions are implemented behind the PMLs to minimize artificial reflections. A periodic port is used to excite the structure with an incident wave. To compute the normalized transmitted power spectra, the transmitted electric field is integrated over a two-dimensional plane located in the air region beyond the cavity. Simulations are performed for various magnetic field strengths $B$ by importing the corresponding permittivity of the magnetoplasma at each $B$, as defined in Eq. (6).

### Derivation of the tight-binding model for graphene
Within a tight-binding model, a solid is considered a collection of atoms with electrons well localized around the atoms. Thus, it is convenient to write the matter Hamiltonian of the crystal $\mathcal{H}_m$ as a sum of a Hamiltonian describing an atom $\mathcal{H}_{at}$ and the potential $\delta V(\mathbf{r})$ describing

the rest of the crystal, $\mathcal{H}_m = \mathcal{H}_{at} + \delta V(\mathbf{r})$. We note that $V_{at}$ and $\delta V(\mathbf{r})$ together give the crystal potential, $V_{crys}(\mathbf{r}) = V_{at}(\mathbf{r}) + \delta V(\mathbf{r})$. It is important to mention that for the construction of the tight-binding ansatz,

$$\Psi_{\mathbf{k}}(\mathbf{r}) = a_{\mathbf{k}}\Psi_{\mathbf{k}}^A(\mathbf{r}) + b_{\mathbf{k}}\Psi_{\mathbf{k}}^B(\mathbf{r}) = \sum_{\mathbf{j}} e^{i\mathbf{k}\cdot\mathbf{R_j}}\left[a_{\mathbf{k}}\phi_A(\mathbf{r} - \mathbf{R_j}) + b_{\mathbf{k}}\phi_B(\mathbf{r} - \mathbf{R_j^B})\right],$$

(20)

the localized states of $\mathcal{H}_{at}$ are used[60,62]. Then we have,

$$\mathcal{H}_{\mathbf{k}}^m = \begin{pmatrix} \langle\Psi_{\mathbf{k}}^A|\mathcal{H}_m|\Psi_{\mathbf{k}}^A\rangle & \langle\Psi_{\mathbf{k}}^A|\mathcal{H}_m|\Psi_{\mathbf{k}}^B\rangle \\ \langle\Psi_{\mathbf{k}}^A|\mathcal{H}_m|\Psi_{\mathbf{k}}^B\rangle^* & \langle\Psi_{\mathbf{k}}^B|\mathcal{H}_m|\Psi_{\mathbf{k}}^B\rangle \end{pmatrix}$$

(21)

The diagonal elements $\langle\Psi_{\mathbf{k}}^B|\mathcal{H}_m|\Psi_{\mathbf{k}}^B\rangle$ and $\langle\Psi_{\mathbf{k}}^A|\mathcal{H}_m|\Psi_{\mathbf{k}}^A\rangle$ result in coupling between next-nearest neighbor sites, hence we eliminate them. The off-diagonal elements are computed as,

$$\langle\Psi_{\mathbf{k}}^A|\mathcal{H}_m|\Psi_{\mathbf{k}}^B\rangle = \langle\Psi_{\mathbf{k}}^A|\mathcal{H}_{at}|\Psi_{\mathbf{k}}^B\rangle + \langle\Psi_{\mathbf{k}}^A|\delta V|\Psi_{\mathbf{k}}^B\rangle$$
$$= E_A \underbrace{\langle\Psi_{\mathbf{k}}^A|\Psi_{\mathbf{k}}^B\rangle}_{=0} + \sum_{\mathbf{j},\mathbf{q}} e^{i\mathbf{k}\cdot(\mathbf{R_j}-\mathbf{R_q})}$$
$$\int d^3r \phi_A^*(\mathbf{r}-\mathbf{R_q})\delta V(\mathbf{r})\phi_B(\mathbf{r}-\mathbf{R_j}-\boldsymbol{\delta}_3).$$

(22)

Next we perform the coordinate shift $\mathbf{r} \to \mathbf{r} + \mathbf{R_q}$ and define $\mathbf{R_f} = \mathbf{R_j} - \mathbf{R_q}$, leading to

$$\langle\Psi_{\mathbf{k}}^A|\mathcal{H}_m|\Psi_{\mathbf{k}}^B\rangle = \sum_{\mathbf{f}} e^{i\mathbf{k}\cdot\mathbf{R_f}}\int d^3r \phi_A^*(\mathbf{r})\delta V(\mathbf{r})\phi_B(\mathbf{r}-\mathbf{R_f}-\boldsymbol{\delta}_3)$$
$$= \sum_{\mathbf{f}} e^{i\mathbf{k}\cdot\mathbf{R_f}} t(|\mathbf{R_f}+\boldsymbol{\delta}_3|)$$

(23)

where $t(|\mathbf{R_f}+\boldsymbol{\delta}_3|)$ is the tunneling matrix element due to the potential $\delta V(\mathbf{r})$, which depends only on the distance between different lattice points. Since we take into account only the nearest neighbor tunneling with the vectors $\mathbf{f} = (f_1, f_2) = (0,0)$, $\mathbf{f} = (1,0)$ and $\mathbf{f} = (0,1)$ and assume no strain, all tunneling elements have the same strength $t(|\mathbf{R}_{0,0}+\boldsymbol{\delta}_3|) = t(|\mathbf{R}_{1,0}+\boldsymbol{\delta}_3|) = t(|\mathbf{R}_{0,1}+\boldsymbol{\delta}_3|) \equiv t$ where $t$ can be found by density functional theory (DFT) as $t = 2.8$ eV[63]. Hence, we find $\langle\Psi_{\mathbf{k}}^A|\mathcal{H}_m|\Psi_{\mathbf{k}}^B\rangle = t\left[1 + e^{i\mathbf{k}\cdot\mathbf{a}_1} + e^{i\mathbf{k}\cdot\mathbf{a}_2}\right] = th(\mathbf{k})$.

### DFT inputs for the microscopic theory of graphene coupled to enhanced vacuum fluctuations
The calculations of DFT were performed using Vienna ab-initio Simulation Package[72–74]. The basis set was projector augmented plane waves with an energy of 500 eV. The Perdew-Burke-Ernzerhof functional was chosen to deal with the exchange-correlation interaction. Spin and spin-orbit coupling were not included. In the reciprocal space, a k-mesh of $9 \times 9 \times 1$ was utilized to sample the momentum. To avoid interlayer interaction from the periodic images, a vacuum space larger than 15 Å was incorporated in the directions vertical to the material plane. The geometric structure was relaxed until the force on each atom was smaller than 0.01 eV/Å.

Here we also provide the derivation between the momentum matrix elements in two different bases, Eq. (17) in the main text. The tight-binding Hamiltonian for graphene expressed above in section "Discussion", $\mathcal{H}_{\mathbf{k}}^m$ in terms of the Pauli matrices, takes the form $\mathcal{H}_{\mathbf{k}}^m = t(d_1(\mathbf{k})\sigma_1 + d_2(\mathbf{k})\sigma_2)$ with

$$d_1(\mathbf{k}) = \cos(\mathbf{k}\cdot\mathbf{a}_1) + \cos(\mathbf{k}\cdot\mathbf{a}_2) + 1,$$

(24)

$$d_2(\mathbf{k}) = \sin(\mathbf{k}\cdot\mathbf{a}_1) + \sin(\mathbf{k}\cdot\mathbf{a}_2),$$

(25)

where $\boldsymbol{a}_1$ and $\boldsymbol{a}_2$ are the translation vectors $\boldsymbol{a}_1 = \boldsymbol{\delta}_1 - \boldsymbol{\delta}_3 = \frac{a}{2}(3, \sqrt{3})$ and $\boldsymbol{a}_2 = \boldsymbol{\delta}_2 - \boldsymbol{\delta}_3 = \frac{a}{2}(3, -\sqrt{3})$. It is standard to diagonalize this Hamiltonian: the valence and conduction band energies are $\epsilon_c(\mathbf{k}) = -\epsilon_v(\mathbf{k}) = t\sqrt{d_1^2(\mathbf{k}) + d_2^2(\mathbf{k})} = td(\mathbf{k})$, and the states are read in terms of sublattices A and B

$$|\Psi_{\mathbf{k}}^c\rangle = \frac{1}{\sqrt{2}}\left(|\Psi_{\mathbf{k}}^A\rangle + \frac{d_1(\mathbf{k}) - id_2(\mathbf{k})}{d}|\Psi_{\mathbf{k}}^B\rangle\right), \quad (26)$$

$$|\Psi_{\mathbf{k}}^v\rangle = \frac{1}{\sqrt{2}}\left(-|\Psi_{\mathbf{k}}^A\rangle + \frac{d_1(\mathbf{k}) - id_2(\mathbf{k})}{d}|\Psi_{\mathbf{k}}^B\rangle\right). \quad (27)$$

Hence, the inverse relation is,

$$|\Psi_{\mathbf{k}}^A\rangle = \frac{\sqrt{2}}{2}\left(-|\Psi_{\mathbf{k}}^v\rangle + |\Psi_{\mathbf{k}}^c\rangle\right), \quad (28)$$

$$|\Psi_{\mathbf{k}}^B\rangle = \frac{\sqrt{2}}{2}\frac{d_1(\mathbf{k}) + id_2(\mathbf{k})}{d}\left(|\Psi_{\mathbf{k}}^v\rangle + |\Psi_{\mathbf{k}}^c\rangle\right) \quad (29)$$

The minimal coupling Hamiltonian for graphene, given in the main text, requires us to compute $i\hbar\langle\Psi_{\mathbf{k}}^A|\nabla|\Psi_{\mathbf{k}}^B\rangle$.

$$\langle\Psi_{\mathbf{k}}^A|\nabla|\Psi_{\mathbf{k}}^B\rangle = \frac{\sqrt{2}}{2}\frac{d_1(\mathbf{k}) + id_2(\mathbf{k})}{d(\mathbf{k})}\left(-\langle\Psi_{\mathbf{k}}^v|\nabla|\Psi_{\mathbf{k}}^v\rangle + \langle\Psi_{\mathbf{k}}^c|\nabla|\Psi_{\mathbf{k}}^c\rangle \right.$$
$$\left. -\langle\Psi_{\mathbf{k}}^v|\nabla|\Psi_{\mathbf{k}}^c\rangle + \langle\Psi_{\mathbf{k}}^c|\nabla|\Psi_{\mathbf{k}}^v\rangle\right). \quad (30)$$

The diagonal terms can be obtained analytically,

$$\langle\Psi_{\mathbf{k}}^c|\nabla|\Psi_{\mathbf{k}}^c\rangle = \frac{1}{2}\left(\langle\Psi_{\mathbf{k}}^A| + \frac{d_1(\mathbf{k}) + id_2(\mathbf{k})}{d(\mathbf{k})}\langle\Psi_{\mathbf{k}}^B|\right)\nabla\left(|\Psi_{\mathbf{k}}^A\rangle + \frac{d_1 - id_2}{d}|\Psi_{\mathbf{k}}^B\rangle\right),$$
$$\langle\Psi_{\mathbf{k}}^v|\nabla|\Psi_{\mathbf{k}}^v\rangle = \frac{1}{2}\left(-\langle\Psi_{\mathbf{k}}^A| + \frac{d_1(\mathbf{k}) + id_2(\mathbf{k})}{d(\mathbf{k})}\langle\Psi_{\mathbf{k}}^B|\right)$$
$$\nabla\left(-|\Psi_{\mathbf{k}}^A\rangle + \frac{d_1(\mathbf{k}) - id_2(\mathbf{k})}{d(\mathbf{k})}|\Psi_{\mathbf{k}}^B\rangle\right). \quad (31)$$

We already assumed in the microscopic theory $\langle\Psi_{\mathbf{k}}^A|\nabla|\Psi_{\mathbf{k}}^A\rangle = \langle\Psi_{\mathbf{k}}^B|\nabla|\Psi_{\mathbf{k}}^B\rangle = 0$. Hence,

$$\langle\Psi_{\mathbf{k}}^c|\nabla|\Psi_{\mathbf{k}}^c\rangle = \frac{1}{2}\left(\frac{d_1(\mathbf{k}) - id_2(\mathbf{k})}{d(\mathbf{k})}\langle\Psi_{\mathbf{k}}^A|\nabla|\Psi_{\mathbf{k}}^B\rangle\right.$$
$$\left. + \frac{d_1(\mathbf{k}) + id_2(\mathbf{k})}{d(\mathbf{k})}\langle\Psi_{\mathbf{k}}^B|\nabla|\Psi_{\mathbf{k}}^A\rangle\right),$$
$$\langle\Psi_{\mathbf{k}}^v|\nabla|\Psi_{\mathbf{k}}^v\rangle = -\frac{1}{2}\left(\frac{d_1(\mathbf{k}) - id_2(\mathbf{k})}{d(\mathbf{k})}\langle\Psi_{\mathbf{k}}^A|\nabla|\Psi_{\mathbf{k}}^B\rangle\right.$$
$$\left. + \frac{d_1(\mathbf{k}) + id_2(\mathbf{k})}{d(\mathbf{k})}\langle\Psi_{\mathbf{k}}^B|\nabla|\Psi_{\mathbf{k}}^A\rangle\right). \quad (32)$$

Substituting (32) in (30), we obtain

$$\left(1 - \frac{\sqrt{2}}{2}\right)\langle\Psi_{\mathbf{k}}^A|\nabla|\Psi_{\mathbf{k}}^B\rangle = \frac{\sqrt{2}}{2}\left(\frac{d_1(\mathbf{k}) + id_2(\mathbf{k})}{d(\mathbf{k})}\right)^2\langle\Psi_{\mathbf{k}}^B|\nabla|\Psi_{\mathbf{k}}^A\rangle$$
$$ - i\sqrt{2}\frac{d_1(\mathbf{k}) + id_2(\mathbf{k})}{d(\mathbf{k})}\mathrm{Im}\langle\Psi_{\mathbf{k}}^v|\nabla|\Psi_{\mathbf{k}}^c\rangle. \quad (33)$$

Also note

$$\left(1 - \frac{\sqrt{2}}{2}\right)\langle\Psi_{\mathbf{k}}^B|\nabla|\Psi_{\mathbf{k}}^A\rangle = \frac{\sqrt{2}}{2}\left(\frac{d_1(\mathbf{k}) - id_2(\mathbf{k})}{d(\mathbf{k})}\right)^2\langle\Psi_{\mathbf{k}}^A|\nabla|\Psi_{\mathbf{k}}^B\rangle$$
$$ + i\sqrt{2}\frac{d_1(\mathbf{k}) - id_2(\mathbf{k})}{d(\mathbf{k})}\mathrm{Im}\langle\Psi_{\mathbf{k}}^v|\nabla|\Psi_{\mathbf{k}}^c\rangle. \quad (34)$$

Solving these two equations, we find

$$\langle\Psi_{\mathbf{k}}^A|\nabla|\Psi_{\mathbf{k}}^B\rangle = -i\sqrt{2}\frac{d_1(\mathbf{k}) + id_2(\mathbf{k})}{d(\mathbf{k})}\mathrm{Im}\langle\Psi_{\mathbf{k}}^v|\nabla|\Psi_{\mathbf{k}}^c\rangle. \quad (35)$$

Going back to the momentum units,

$$i\hbar\langle\Psi_{\mathbf{k}}^A|\nabla|\Psi_{\mathbf{k}}^B\rangle = -\sqrt{2}\frac{d_1(\mathbf{k}) + id_2(\mathbf{k})}{d(\mathbf{k})}\mathrm{Re}\left[i\hbar\langle\Psi_{\mathbf{k}}^v|\nabla|\Psi_{\mathbf{k}}^c\rangle\right]. \quad (36)$$

## Data availability

The numerical simulation data used to generate the plots are available at the Rice Research Repository (R-3) at https://hdl.handle.net/1911/118335.

## Code availability

The codes used in this study are available from the corresponding author upon request.

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

## Acknowledgements

The authors thank David Hagenmüller and Xiangfeng Wang for useful discussions. J.K. acknowledges support from the U.S. Army Research Office (through award no. W911NF2110157), the Gordon and Betty Moore Foundation (through grant no. 11520), the W.M. Keck Foundation (through award no. 995764), and the Robert A. Welch Foundation (through grant no. C-1509). V.R. and C.B.D. acknowledge support from the National Science Foundation grant for ITAMP at Harvard University [award no: 2116679].

## Author contributions

F.T., V.R., C.B.D. and J.K. conceptualized the project. F.T. designed the cavity and performed numerical simulations with input from S.S., A.B., A.A., V.R. and C.B.D. developed the microscopic model and performed calculations. Z.S. and D.M.W. performed DFT calculations. V.R., C.B.D., A.B. and J.K. supervised the project. F.T., V.R., C.B.D. and J.K. prepared the manuscript with inputs from all authors.

## Competing interests

The authors declare no competing interests.
