## [Transparent Peer Review file · Nature Communications]

Terahertz chiral photonic-crystal cavities for Dirac gap engineering in graphene

Corresponding Author: Professor Junichiro Kono

Version 0:

Reviewer comments:

Reviewer #1

(Remarks to the Author)

This work numerically investigates a terahertz chiral photonic crystal cavity with broken time-reversal symmetry. By utilizing magnetoplasma in a lightly doped semiconductor, the authors demonstrate non-reciprocal transmission of chiral light. While the results are technically sound and consistent, the primary concern lies in the limited novelty of the research. Although the findings are valid, they may not constitute a substantial advancement in the field, raising questions about the work's suitability for publication in Nature Communications.

Major Comment: Limited Novelty of Results

1.Commonality of Proposed Cavity Design

While the authors claim that their optimized cavity achieves a high-quality factor under a small external magnetic field, the proposed 1D photonic crystal cavity designs (Figs. 3c-3f) are not novel. Similar structures have been widely studied, and the comparison between different cavities appears repetitive rather than introducing new insights.

2.Predictable Non-Reciprocal Propagation

The introduction of gyrotropic materials (replacing some silicon layers) to achieve non-reciprocal light propagation is an expected outcome. This approach has been previously explored in various contexts, and the results do not provide groundbreaking advancements.

3.Numerical Nature of the Work

As this is entirely a numerical study, the lack of experimental validation diminishes its impact. Pioneering research expected in Nature Communications typically includes experimental demonstrations to support theoretical or numerical findings, especially for a well-established structure such as this one.

Reviewer #2

(Remarks to the Author)

In this theoretical work, the authors propose a new approach towards THz chiral photonic crystal cavities. They work with doped semiconductors and thus achieve high quality factors. The time-reversal-symmetry-breaking chiral resonances they accomplish are then applied to a thin graphene layer, where strong light-matter interaction produces a measurable energy gap in the graphene of 1 meV.

Chirality and time-reversal symmetry breaking are topics of great current interest because they permit novel properties in topological and quantum materials. Unidirectional propagation is one of these, and also considered by the authors. With their superior design, the chiral cavities permit effective coupling to such states, and a range of cavity geometries is presented that gives deeper insight into the new design concept. With all this, I recommend publication in Nature Communications. The authors might like to refer to some old work, where the idea of symmetry-engineered geometries for THz propagation was initially presented. See A. Figotin, I. Vitebskiy, Phys. Rev. B 67, 165210 (2003), for example.

Reviewer #3

(Remarks to the Author)

The paper by F. Tay et al deals with the topic of strong light-matter interaction and focuses on the design of an electromagnetic cavity supporting only one circularly polarized mode. This breaks the time reversal symmetry which can

lead to new states of matter if such resonator is coupled to a fermionic system.

The introduction is clear and well-written. The electrodynamic part is good understandable as well. The aim is clear and the methods used suits the goal. The minor comment on this part is the usage of the COMSOL finite-element method solver for the current problem. In my opinion the transfer matrix method is sufficient and more appropriate for the one-dimensional problem concerned in the current manuscript. The references cover the previous work on the topic and provide enough overview to the state of the art. The minor drawback is the lack of citations to the original work, such as Drude model, etc, with biased emphasis on the very recent publications.

The second part of the manuscript devoted to the light-matter interaction looks plausible as well, although I have less hands-on experience in the topic. All model calculations are based either on the established basic concepts or the well accepted current understanding of the problem in the community. The conclusions are well justified and the numerical estimates are made within reasonable assumptions. However, there is one effect that is completely missing in the discussion, which I address separately below.

Overall, the manuscript is a solid well-structured theoretical work. Here I am not sure that the purely theoretical paper is worth publishing in Nature Communications. The novel concepts are very important in the scientific community and are valuable by itself. The present work however builds up upon previous work. The way of achieving the time-reversal broken cavity modes is very promising, but there always could be unforeseen conditions that break the assumptions of the models and prevent the experimental realization of the idea. Depending on how the influence of static magnetic field on the fermionic system (see below) is considered, the paper can be still a significant contribution to the field.

The main concern is the influence of the static magnetic field on the electronic system (graphene). The static magnetic field is necessary to break the time reversal symmetry of the cavity mode. It should be present on all InSb layers placed on both sides of the defect Si layer. Therefore, the magnetic field will be present on the Si defect layer as well in a realistic experimental realization. However, the same magnetic field will break the time reversal symmetry of the electronic system by itself. The most trivial effect is the formation of Landau levels in two-dimensional electron system (graphene). This, or other possible effects are not even mentioned in the manuscript. In my opinion, this is a major lacking part that should be covered in the manuscript.

Version 1:

Reviewer comments:

Reviewer #1

(Remarks to the Author)

I am very grateful to the authors for their response to my comments, and I acknowledge their efforts. However, regarding this manuscript, I still maintain my original opinion. The paper discusses the defect modes in one-dimensional photonic crystal structures. Even though the defect mode is described by the authors as "the integration of magnetic lightly doped semiconductors into 1D-PCCs," I do not believe it constitutes a significant contribution to the field. In addition, the non-reciprocal propagation under time-reversal symmetry is not surprising due to the introduction of magnetic materials. Therefore, I maintain my review opinion and oppose the publication of this article in Nature Communications.

Reviewer #3

(Remarks to the Author)

The authors have taken the comments of the referees seriously and substantially reworked the manuscript. I appreciate the comparison of transmittance-matrix method with the finite-element calculations. The literature overview has a good broad coverage as well. The clarification on the novelty of the purely theoretical approach is also plausible.

The answer to the main concern of the influence of static magnetic field on the electronic system itself is best summarized in the Discussion and Outlook section of the revised manuscript:

"...At $B = 0.2$ T, ... ,the energy spacing between the Landau levels $\nu = 1$ and $\nu = 0$ is on the order of 10 meV.

...

Within this theoretical treatment without a magnetic field, we estimated a cavity-induced gap of ~ 1 meV..."

In my opinion, this is a negative result of the present study. The direct effect of static magnetic field is larger than the studied cavity-induced gap.

However, I do not like the present trend of rejecting papers with negative results from high-impact journals. Therefore, I recommend publishing the manuscript in Nature Communications in its present honest form.

Response Letter to Reviewers

Each comment is in blue, quoted in Italics, followed by our response. Major updates in the main text and Supplementary Information are highlighted in red fonts.

RESPONSE TO THE 1st REVIEWER:

This work numerically investigates a terahertz chiral photonic crystal cavity with broken time-reversal symmetry. By utilizing magnetoplasma in a lightly doped semiconductor, the authors demonstrate non-reciprocal transmission of chiral light. While the results are technically sound and consistent, the primary concern lies in the limited novelty of the research. Although the findings are valid, they may not constitute a substantial advancement in the field, raising questions about the work's suitability for publication in Nature Communications.

We thank the referee for their critical comments, which helped us to improve our manuscript. We address the referee's concerns below on a point-by-point basis.

1. Commonality of Proposed Cavity Design

While the authors claim that their optimized cavity achieves a high-quality factor under a small external magnetic field, the proposed 1D photonic crystal cavity designs (Figs. 3c-3f) are not novel. Similar structures have been widely studied, and the comparison between different cavities appears repetitive rather than introducing new insights.

We respectfully disagree with the reviewer's comment. While nonmagnetic 1D photonic-crystal cavities (PCCs) with a central defect layer has been widely studied, the integration of magnetic lightly doped semiconductors into 1D-PCCs has not been explored to the best of our knowledge. Most importantly, a key challenge in designing such cavities is ensuring that the carrier density, material choice, and the number of lightly doped semiconductor layers do not degrade the cavity's quality factor, which is a nontrivial problem.

Our extensive analysis (Fig. 3c–3f) systematically evaluated the optimal configuration for chiral PCCs. For instance, previous studies [K.W. Lee *et al.*, *Opt. Mater. Express* 4, 2542 (2014), A.H. Aly *et al.*, *Int. J. Mod. Phys. B* 31, 1750239 (2017), & T. Li *et al.*, *Opt. Mater.* 121, 111583 (2021)] considered magnetic PCCs by replacing the central defect layer with a magnetized plasma or a Weyl semimetal. However, our analysis showed that chiral cavity performance is highly sensitive to the losses in the central defect layer. Hence, to mitigate this, our design replaces one of the constituent media with a lightly doped semiconductor while keeping the defect layer dielectric, ensuring both a high-quality factor and a strong electric field enhancement.

In addition, we carefully chose the carrier density and thickness of the lightly doped semiconductor to achieve sufficient suppression of one handedness (ellipticity ~ 1) while maintaining a dielectric

behavior for the opposite handedness. Furthermore, the low effective mass of electrons in InSb ensures that the cyclotron resonance of the magnetoplasma occurs within the terahertz frequency range in a relatively weak magnetic field, allowing for efficient circularly polarized cavity fields at low magnetic fields. These design details are crucial for exploring predicted phenomena in chiral cavity quantum electrodynamics.

We have revised the Introduction section in the revised manuscript (the red highlighted text) to clarify these novel aspects. We have also added a sentence on Page 11 to explicitly distinguish our work from prior studies:

“Notably, despite the defect layer being nonmagnetic in Design II, the value of $\eta(z)$ at z_{\max} (the defect layer surface) still reaches 1. This configuration differs from the conventional structure explored in previous studies, which typically incorporate a magnetic defect layer within a photonic crystal [50–52].”

2. Predictable Non-Reciprocal Propagation

The introduction of gyrotropic materials (replacing some silicon layers) to achieve non-reciprocal light propagation is an expected outcome. This approach has been previously explored in various contexts, and the results do not provide groundbreaking advancements.

We would like to emphasize that while non-reciprocal light propagation using gyrotropic materials in PCCs have been explored in the context of unidirectional devices and enhancement of magneto-optical effects [e.g., A. Figotin, I. Vitebskiy, *Phys. Rev. B* 67, 165210 (2003), M. Inoue, *et al.*, *J. Phys. D: Appl. Phys.* 39, R151–161 (2006)], it has never been investigated in the framework of chiral PCCs with broken time-reversal symmetry. Specifically, our current work explores a distinct magnetic resonance (i.e., the magnetoplasma in low-doped InSb), systematically compares different cavity configurations, and analyzes the ellipticity of the cavity profiles. These aspects are carefully designed to achieve a high-quality-factor photonic cavity with uniform and near-unity ellipticity as well as broken time-reversal symmetry at relatively low magnetic fields. No prior studies have made such achievements.

3. Numerical Nature of the Work

As this is entirely a numerical study, the lack of experimental validation diminishes its impact. Pioneering research expected in Nature Communications typically includes experimental demonstrations to support theoretical or numerical findings, especially for a well-established structure such as this one.

We respectfully disagree with the referee. Nature Communications publishes theoretical papers, and the scope of the journal is not limited to theory-experiment collaborations. Furthermore, this work is not an entirely numerical study. Our theory incorporates an analytical cavity QED model for graphene, *ab initio* simulations of Maxwell’s equations to obtain the exact field profile for the desired chiral cavity, and *ab initio* density functional theory (DFT) to calculate the transition matrix elements of monolayer graphene. The combination of these methods enabled us to determine the exact light–matter interaction strength and provide a realistic estimate for the cavity-

induced topological Dirac gap. To the best of our knowledge, this has not been done before, and we believe that it is a significant step towards realizing cavity-dressed condensed matter phases with broken time-reversal symmetry. In fact, due to the combination of these different theoretical techniques, we discovered, and now predict, an enhancement in the light–matter interaction around the Dirac nodes. This importantly implies that even for a cavity that is not sub-wavelength, an observable effect might still be obtained solely due to the band structure of the underlying material.

To highlight all these different elements entering our theory, we have included an illustration in Fig. 1 and have expanded the discussion of this important point in the Introduction of the manuscript on Page 3. We now also emphasize the Dirac node enhancement of the light–matter interaction in both the abstract and the Introduction. Please refer to the highlighted parts in the revised manuscript.

RESPONSE TO THE 2nd REVIEWER:

In this theoretical work, the authors propose a new approach towards THz chiral photonic crystal cavities. They work with doped semiconductors and thus achieve high quality factors. The time-reversal-symmetry-breaking chiral resonances they accomplish are then applied to a thin graphene layer, where strong light-matter interaction produces a measurable energy gap in the graphene of 1 meV.

Chirality and time-reversal symmetry breaking are topics of great current interest because they permit novel properties in topological and quantum materials. Unidirectional propagation is one of these, and also considered by the authors. With their superior design, the chiral cavities permit effective coupling to such states, and a range of cavity geometries is presented that gives deeper insight into the new design concept. With all this, I recommend publication in Nature Communications. The authors might like to refer to some old work, where the idea of symmetry-engineered geometries for THz propagation was initially presented. See A. Figotin, I. Vitebskiy, Phys. Rev. B 67, 165210 (2003), for example.

We are delighted to hear the Referee’s recommendation of publication of our manuscript in *Nature Communications*. We have now included earlier studies of magnetic photonic crystals that aim to achieve the unidirectional propagation of light and enhanced magneto-optical effects on Page 3:

“Similar designs, including Fabry–Perot and Bragg cavities incorporating Faraday rotation materials, have been explored earlier [22–24] to enhance magneto-optic effects and achieve unidirectional transmission.”

RESPONSE TO THE 3rd REVIEWER:

The paper by F. Tay et al deals with the topic of strong light-matter interaction and focuses on the design of an electromagnetic cavity supporting only one circularly polarized mode. This breaks the

time reversal symmetry which can lead to new states of matter if such resonator is coupled to a fermionic system.

The introduction is clear and well-written. The electrodynamic part is good understandable as well. The aim is clear and the methods used suits the goal. The minor comment on this part is the usage of the COMSOL finite-element method solver for the current problem. In my opinion the transfer matrix method is sufficient and more appropriate for the one-dimensional problem concerned in the current manuscript. The references cover the previous work on the topic and provide enough overview to the state of the art. The minor drawback is the lack of citations to the original work, such as Drude model, etc, with biased emphasis on the very recent publications.

We sincerely appreciate the Referee’s positive feedback and constructive criticism, which have helped us to improve our manuscript. We fully agree that the transfer matrix method (TMM) is a well-established and efficient tool for one-dimensional problems. Indeed, our findings can be reproduced using the TMM, particularly in the case of circularly polarized light, where the 3x3 permittivity tensor of the magnetoplasma becomes diagonal in the circular basis. As shown in Fig. R1, the transmittance spectra obtained using TMM and COMSOL simulations are in excellent agreement.

However, for linearly polarized incident light, extensions of the typically isotropic TMM to an anisotropic version are required to account for the off-diagonal elements of the permittivity tensor. By contrast, COMSOL allows us to directly input the full 3x3 permittivity tensor without additional modifications. In addition, COMSOL provides a more versatile framework for future studies, enabling the exploration of more complex structures, such as incorporating a metasurface array on the defect layer surface to further reduce the mode volume.

Therefore, while an anisotropic TMM can be used for these calculations, we employed COMSOL due to its convenience in incorporating a full 3x3 permittivity tensor and its adaptability for potential extensions of our study. To clarify this, we added a section to the Supplementary Information.

Fig. R1 Normalized power spectra for incident light with the “-” polarization at $B = 0.212$ T calculated using the transfer matrix method (TMM) and COMSOL Multiphysics.

In addition, we now include citations to earlier, pioneering studies on the Drude model and magnetoplasmas in semiconductors, as suggested by the Referee. The citations have been added on Page 7:

“Lightly doped n -InSb contains a low-density plasma with a plasma frequency in the THz frequency range, and a magnetoplasma with nonreciprocity properties is formed when an external magnetic field, B , is applied [37–41]. The complex permittivity of the magnetoplasma at B is given by a gyrotropic permittivity tensor [42–46] ...”

We have also included references to earlier studies on related topics in our manuscript. For example, citations on magnetic photonic crystals have been added on Page 2:

“Similar designs, including Fabry–Perot and Bragg cavities incorporating Faraday rotation materials, have been explored earlier [22–24] to enhance magneto-optic effects and achieve unidirectional transmission.”

The second part of the manuscript devoted to the light-matter interaction looks plausible as well, although I have less hands-on experience in the topic. All model calculations are based either on the established basic concepts or the well accepted current understanding of the problem in the community. The conclusions are well justified and the numerical estimates are made within reasonable assumptions. However, there is one effect that is completely missing in the discussion, which I address separately below.

Overall, the manuscript is a solid well-structured theoretical work. Here I am not sure that the purely theoretical paper is worth publishing in Nature Communications. The novel concepts are very important in the scientific community and are valuable by itself. The present work however builds up upon previous work. The way of achieving the time-reversal broken cavity modes is very promising, but there always could be unforeseen conditions that break the assumptions of the models and prevent the experimental realization of the idea. Depending on how the influence of static magnetic field on the fermionic system (see below) is considered, the paper can be still a significant contribution to the field.

We thank the Referee for their constructive feedback. Indeed, our work is theoretical, bringing multiple different methods together: an analytical cavity QED model for graphene, *ab initio* simulations of Maxwell’s equations to obtain the exact field profile for the desired chiral cavity, and *ab initio* density functional theory (DFT) to calculate the transition matrix elements of monolayer graphene. The combination of these methods enables us to determine the exact light–matter interaction strength and provide a realistic estimate for the cavity-induced topological Dirac gap. In fact, due to the combination of these different theoretical techniques, we find and predict an enhancement in light–matter interaction around the Dirac nodes. This importantly implies that even for a cavity that is not sub-wavelength, an observable effect might still be obtained solely due to the band structure of the underlying material. To emphasize how we bring together these different

theoretical models to obtain a realistic gap estimate, we now appended a subfigure to Fig. 1 summarizing all methods used in the manuscript.

We also thank the referee for their criticism, which we address below.

The main concern is the influence of the static magnetic field on the electronic system (graphene). The static magnetic field is necessary to break the time reversal symmetry of the cavity mode. It should be present on all InSb layers placed on both sides of the defect Si layer. Therefore, the magnetic field will be present on the Si defect layer as well in a realistic experimental realization. However, the same magnetic field will break the time-reversal symmetry of the electronic system by itself. The most trivial effect is the formation of Landau levels in a two-dimensional electron system (graphene). This, or other possible effects, are not even mentioned in the manuscript. In my opinion, this is a major lacking part that should be covered in the manuscript.

Indeed, the Referee is correct that in the current design, even though the required magnetic field is small, it will inadvertently affect the material under study. In this study, we have chosen to neglect the effects of the magnetic field on graphene, including the effect of Landau quantization, to simplify the analysis. The treatment of fully filled graphene Landau levels coupled to a cavity field differs from the existing theories utilizing the Hopfield model for GaAs Landau levels coupled with cavity modes, due to the anharmonic gaps between graphene Landau levels. While addressing this effect is beyond the scope of the present study, we are fully aware of the importance of including magnetic field effects in future theoretical developments to provide a more complete understanding and better interpret experimental results.

We would like to emphasize, however, the significant advancement made by our current design in substantially reducing the required magnetic field to achieve chiral cavity modes with broken time-reversal symmetry, specifically by almost two orders of magnitude compared to previous studies [Andberger, J., *et al.*, *Phys. Rev. B* 109, L161302 (2024), Suárez-Forero, D. G., *et al.*, *Sci. Adv.* 10, eadr5904 (2024)]. This reduction is crucial, as it brings us closer to the goal of realizing a magnetic-field-free chiral cavity, which remains an open challenge in the field.

In addition, the current cavity design, with its required low magnetic field, can still be very useful in exploring *cavity-enhanced effects*, rather than cavity-induced effects. In particular, while the time-reversal symmetry in graphene is also broken by the applied magnetic field, which differs from the original proposals of cavity-induced quantum anomalous Hall effect, it remains an intriguing open question whether the chiral cavity can modify phenomena induced by the static magnetic field, e.g., shifting or splitting Landau levels, particularly the $\nu = 0$ state. In this sense, our current cavity design provides an experimental platform to explore such cavity-enhanced effects at low magnetic fields. Since cavity-enhanced effects are often more accessible experimentally than purely cavity-induced phenomena, our work represents an important step in this direction and will also motivate advancements in theory.

Furthermore, potential improvements to the cavity design can also be made by material design, such as replacing InSb slabs with alternative materials that require even weaker magnetic fields to achieve chirality to further minimize the magnetic field's effect on the central plate region.

We sincerely thank the reviewer for highlighting these important considerations. We have revised the manuscript to explicitly clarify the approximation of neglecting the magnetic field's influence on graphene in our theoretical model. Moreover, we include an extensive discussion of these points in the Discussion and Outlook section of our manuscript, highlighting both the current study's significance and the future directions it enables.